# Study of the Floristic, Morphological, and Genetic (atpF–atpH, Internal Transcribed Spacer (ITS), matK, psbK–psbI, rbcL, and trnH–psbA) Differences in *Crataegus ambigua* Populations in Mangistau (Kazakhstan)

**DOI:** 10.3390/plants13121591

**Published:** 2024-06-07

**Authors:** Akzhunis Imanbayeva, Nurzhaugan Duisenova, Aidyn Orazov, Meruert Sagyndykova, Ivan Belozerov, Ainur Tuyakova

**Affiliations:** Laboratory of Natural Flora and Dendrology, Mangyshlak Experimental Botanical Garden, Aktau 130000, Kazakhstan; imanbayeva_a@mebs.kz (A.I.); duisenova_n@mebs.kz (N.D.); m.sagyndykova@mail.ru (M.S.); bif17@mail.ru (I.B.); ainura_kosai@mail.ru (A.T.)

**Keywords:** *Crataegus ambigua*, arid territory, diversity, genetic markers, phytosociology

## Abstract

This article studies the morphological parameters of vegetative and generative organs of different age groups of *Crataegus ambigua* from four populations in Western Karatau (Mangistau region, Kazakhstan). In this study, we examined four populations: Sultan Epe, Karakozaiym, Emdikorgan, and Samal, all located in various gorges of Western Karatau. Several phylogenetic inference methods were applied, using six genetic markers to reconstruct the evolutionary relationships between these populations: atpF–atpH, internal transcribed spacer (ITS), matK, psbK–psbI, rbcL, and trnH–psbA. We also used a statistical analysis of plants’ vegetative and generative organs for three age groups (virgin, young, and adult generative). According to the age structure, Samal has a high concentration of young generative plants (42.3%) and adult generative plants (30.9%). Morphological analysis showed the significance of the parameters of the generative organs and separated the Samal population into a separate group according to the primary principal component analysis (PCoA) coordinates. The results of the floristic analysis showed that the Samal populations have a high concentration of species diversity. Comparative dendrograms using UPGMA (unweighted pair group method with arithmetic mean) showed that information gleaned from genetic markers and the psbK–psbI region can be used to determine the difference between the fourth Samal population and the other three.

## 1. Introduction

The genus *Crataegus* L. (hawthorn) is one of the ancient and rich genera of species belonging to the family Rosaceae Juss [1,2,3,4]. Species of the genus *Crataegus* L. were found on Earth in the *Cretaceous* period of the Mesozoic era, along with magnolias (*Magnolia* L.), bay trees (*Laurus* L.), tulip trees (*Liriodendron* L.), plane trees (*Platanus* L.), and other broad-leaved genera [5]. Representatives of the hawthorn genus spread widely in the Tertiary period [6,7]. During the Quaternary period, many hawthorns died due to glaciation, and some began migrating south [8]. After glaciation, many species began migrating north again [9,10]. These were the ancestors of the modern Eurasian (*Crataegus laevigata* (Poir.) DC.) and several American dissected-leaf and small-thorned hawthorns [11]. *Crataegus* is very polymorphic and has about 300 species [12]. Some researchers have estimated the diversity of hawthorn to be 1000 species, as hawthorn species easily cross with each other and take many different individual and hybrid forms [13]. According to various taxonomic systems, this genus includes 1250 species, of which 1125 are American (also according to Krussmaim [14]). A total of 90 species grow in Europe and Asia, and up to 800 grow in America; some 39 species grow in the territory of the former Union of Soviet Socialist Republics (USSR) [15]. In Kazakhstan, according to the Flora of Kazakhstan, there are seven species [16], and according to M.S. Baitenov, there are nine species [17]. One of the few representatives of the tree flora of Mangistau is the dubious hawthorn (*C. ambigua* C. A. Mey. ex A. K. Becker) [18].

*C. ambigua* grows as a shrub or small tree 3–4 m in height. The shoots are non-thorny, red–brown, and covered with skin. The leaves are broadly ovate and lobed, and the petioles are short. The inflorescence produces from 12 to 20 flowers. The fruits are purple–black, spherical, and broadly elliptical. In the northern hemisphere, this plant blooms in May, bears fruit in July and August, and reproduces by seeds. Autumn sowing with freshly harvested seeds gives good results in the seed germination process. *C. ambigua* plants are of economic value as ornamental plants and as producers of biologically active substances. The plant is mesoxerophytic and endemic to Western Kazakhstan. It is listed in the Red Book of Kazakhstan [19] and in the Catalog of Rare and Endangered Plant Species of the Mangistau Region [20] as a rare species in the republic. Due to the polymorphic nature of the hawthorn, there are two interpretations of its species. In several works, the name *C. ambigua* is retained for the Mangyshlak hawthorn [21], while other authors [22,23,24,25] recognise it as an independent species of *C. trancaspica* (the trans-Caspian hawthorn). This plant grows mainly along the bottom of ravines, chalk, and gypsum gorges and in damp places. It is found in the Western and Eastern Karatau mountains, the western part of Northern Aktau (Emdy Ridge), the Tyubkaragan Peninsula, and Northern Ustyurt [26]. Natural populations of dubious hawthorn are found in the mountains of Western and Eastern Karatau, the western part of Northern Aktau, the Tyubkaragan Peninsula, Northern Ustyurt, and outside Kazakhstan in the southeast of the European part of Russia. It grows individually or in small groups on rocky slopes and at the bottom of wet gorges [27,28]. The number of dubious hawthorn individuals tends to decrease due to the harvesting of hawthorn wood for fuel and overgrazing by livestock. The plant is characterised by high salt, heat, and drought resistance. The authors of the Catalog of Rare and Endangered plant species of the Mangistau region (Red Book) (2006) recommended the organisation of a reserve in the mountains of Mangyshlak Karatau for the protection of hawthorn and several other rare plants [29].

Carrying out a comprehensive assessment of the population genetic material of *C. ambigua* from natural habitats is necessary to identify donors of valuable economic and biological traits. Studying the same forms under natural and introduced conditions allows us to identify plants’ phenotypic and genotypic variability. Currently, the amount of research on genetic characteristics could be more significant. However, there are studies on the centre of genetic diversity of the section *Crataegus*, which extends from Turkey to Iran [30]. Modern advances in molecular biology make it possible to carry out genomic typing of plants from *C. ambigua* populations based on data on the primary DNA structure [31,32,33,34], allowing standardising the procedure for identifying *Crataegus* species using reference DNA loci [35,36,37,38,39].

It is known that ITS DNA loci are located between the structural genes of ribosomal RNA 18S, 5.8S, and 26S, forming a single cluster of nuclear genes that are organised in the form of tandem DNA repeats. Each ribosomal gene cluster consists of a transcribed region (genes 18S, 5.8S, and 26S); internal transcribed spacers located on either side of 5.8S, named ITS1 and ITS2, respectively; and flanking external transcribed spacers, ETS1 and ETS2 [40]. ITS markers are phylogenetic markers that classify plants at different taxonomic levels, including genus, species, and subspecies [41,42].

The primary sequences of the genes rpoB, rpoC1, rbcL, matK, psbK–psbI, trnH–psbA, atpF, and atpH are used in determining the origin and relationships of *Crataegus* plants [43]. The non-coding regions of the rpoB, rpoC1, rbcL, matK, psbK–psbI, trnH–psbA, atpF, and atpH genes are subject to significant mutational events and occupy most of the plastome. In contrast, the coding regions of these genes evolve slowly and are highly conserved between species. Determination of the primary structure of intergenic spacers of these genes and introns has great practical significance in the phylogenetic study of plants, making a solid contribution to the processes of introducing and selecting economically valuable plants. The main goals of this study are to study new populations in the Mangistau region based on morphological and genetic parameters and to identify the main differences between *C. ambigua* populations and methods for their conservation [44]. These studies may also be referenced in the selection and introduction of natural populations and varieties of *C. ambigua*, both of which require advanced molecular biological methods to mark original DNA lines for selection. Selecting and assessing promising genotypes of collection funds using DNA barcoding of constitutive loci from both genomic and cytoplasmic DNA is advisable. The DNA barcode allows the identification of *C. ambigua* plants used for selection and introduction at each stage of the breeding process.

## 2. Methods

### 2.1. Study Area

The study area is located in the western part of the Karatau mountain system of the Mangystau region of Kazakhstan, which has a relatively high concentration of biologically diverse arid plant species, including valuable fruit and berry trees. The Mangistau region is located west of Kazakhstan, in a desert zone with various mountain systems. Karatau is a large and significant mountain system with a length of 110 km from west to east and a width of 12 km. It is divided into Western and Eastern Mangystau. The highest peak is Mount Beshoky (552 m) in Western Mangystau. Ravines and potholes cut the foothills of the mountain ranges. The western part of Karatau has several mountain gorges, forming isolated ecosystems. This mountainous terrain formed during the Permian period from metamorphic limestones, sandstones, and conglomerates. These gorges are suitable for grazing [45]. The climate is sharply continental and extremely dry, with hot summers and moderately cold winters. This territory can be classified as an arid zone, featuring unfavourable conditions for most species of vascular plants; this, in turn, increases interest in this territory and its natural populations [46].

### 2.2. Plant Materials

Plant samples from four populations of *C. ambigua* from Western Karatau, Mangystau were studied as a basis for studying morphological and genetic parameters. Homogeneous populations were selected for the study. These populations were located in isolated gorges. Pop 1 is made up of Sultan Epe in the Sultan Epe Gorge (Tyubkaragan Peninsula), Pop 2, Karakozaiym, is in the Karakozayim Gorge (Tyubkaragan Peninsula), Pop 3, Emdikorgan, is in the Emdikorgan Gorge (Northern Aktau Ridge), and Pop 4, Samal, is in the Samal Gorge (Western Karatau Ridge). Table 1 shows the central coordinates of the populations’ locations.

Each population has unique conditions. The highest population is Pop 4, Samal, 247 m above sea level, and the lowest population is Pop 3, Emdikorgan, 35 m above sea level. Figure 1 shows the location of the four study populations and photographs under natural conditions.

As plant materials for the study of morphological indicators, ten individual reference plants were selected from natural populations of different ages (virginal, young generative, and adult generative), characterised as low trees and shrubs with brownish–grey bark and old trees with reddish–brown shoots, grey coating, and branching up to the sixth order. Often light brown, young shoots are covered with axillary spines or are sometimes bare; leaves are alternately pinnate or serrated. The morphological parameters of the vegetative and generative organs of hawthorns of different ages were also analysed (Figure 2).

Additionally, when studying morphological parameters, the age composition of the populations was considered and three groups of *C. ambigua* plants of different ages were identified from the four populations: virgin (V), young generative (YG), and adult generative (AG). The following parameters of the vegetative organs of plants were studied: plant height, cm (PH); plant crown diameter, cm (CD); trunk height, cm (TH); trunk diameter, cm (TD); spike size, cm (SS); leaf length, cm (LL); leaf width, cm (SW); leaf petiole length, cm (LPL); and leaf area, cm (LA). The following parameters of the generative organs were also studied: inflorescence diameter, cm (ID); number of flowers on one inflorescence, pcs (NFI); number of inflorescences on one branch, pcs (NIB); pedicel length, mm (PL); flower diameter, mm (FD); fruit weight (FM); fruit length (FL); fruit width (FW); number of seeds (NS); and seed mass (SM).

### 2.3. Floristic Analysis

A population analysis of the floristic properties of plants during ontogenesis was undertaken using the method (based on the population concept in phytocoenology and solving its problems) of O. V. Smirnova et al. [47]. At the same time, the names of plant species were verified against the Flora of Kazakhstan [28], Illustrated Key [48], Identifier of Plants of Central Asia [49], and the International Index of Names of Electronic Databases [50]. The classification of life forms of species included in the population was assessed using the I.G. Serebryakov method [51]. Traditional geobotanical survey methods use ecological and morphological indicators to describe the populations, such as establishing and marking a trial plot, filling out a phytocoenoses description form, and other basic methods. Individuals of *C. ambigua* were counted by age group, and their floristic composition was described [52]. Plant nomenclature was determined according to POWO [53]. When describing ontogeny, we used the method of Komarov et al. [54]. The population type was determined using the technique of S.V. Fedorov [55]. The following age groups were considered: young individuals (of root or seed origin), virginal (large individuals but not reaching the generative period), young generative, and adult generative. Seedlings and senile individuals were not identified in the natural population at the time of this study.

### 2.4. Genetic Analysis

Fresh leaves of *C. ambigua* plants from the four populations studied were used for DNA extraction. Ten reference plant samples were selected from each population for each marker to study genetic diversity. The Shannon and Simpson Diversity indices were used to study genetic diversity. Total DNA was isolated from ground leaf powder using the cetyltrimethylammonium bromide (CTAB) protocol with double chloroform purification [56]. DNA quality and concentration were assessed using a NanoDrop 2000 spectrophotometer (Thermo Fisher Scientific, Waltham, MA, USA) and 1% agarose gel electrophoresis. The DNA concentration was normalised to the working concentration for further analysis. The buffers were prepared according to existing guidelines [57]. To isolate DNA, a method using cetyltrimethylammonium bromide was used [58]. DNA sequencing was performed using the Sanger method using a BigDye™ Terminator v3.1 Cycle Sequencing Kit [59]. The data on the primary structure of the DNA fragments under study were analysed through the National Center for Biotechnology Information (NCBI) genetic information database using the BLAST+ 2.15.0 (Basic Local Alignment Search Tool) program [60]. To barcode the plants selected in the study, we selected DNA sequences of the chloroplast genome, such as matK, rbcL, trnH–psbA, psbK–psbI, and atpF–atpH, as well as nuclear DNA fragments ITS1 and ITS2. Table 2 shows the sequences of the primers used in the study.

The resulting DNA sequences were edited and contiguous sequences assembled (from 193 to 570 bp) using the MEGA program, version 5 [61]; the resulting DNA sequence information was then utilised for fundamental evolutionary analyses [62], evolutionary distances, and phylogenetic reconstructions [63]. The calculated Shannon and Simpson indices for each group of *C. ambigua* sequences allow us to assess the nucleotide diversity in each population. The Shannon index considers both the richness (number of unique nucleotides) and the evenness of their distribution.

### 2.5. Statistical Processing

Statistical processing was carried out via a nonparametric Mann–Whitney test to determine the significance of the differences between floristic composition indicators between populations and the differences between the morphological indicators of populations of the studied species. Mathematical processing was carried out according to Kuziev R. and Yuldashev G. et al. [64,65]. Statistical analysis of population dynamics varies depending on the limits imposed, which vary according to the standard deviation of the coefficient of variation, the mean error, the degree of confidence, and the degree of precision. Dendrograms were constructed using PAST 4.03 and the unweighted pair group method with an arithmetic mean algorithm (UPGMA) and Boot N:1000 [66]. When primary data were analysed, correlation coefficients were calculated using the R program for Windows (R version 3.6.0, 2019). A multivariate analysis of variance (MANOVA) was performed during the study, with results including Wilks’ lambda and Pillai trace indicators. The ADONIS assay method was also used. The mean values of the main quantitative traits of the four populations were grouped using the PCoA (principal component analysis) method. PCoA was performed using the Numerical Taxonomy and Multivariate Analysis System version 2.1. (NTSYS-pc) [29].

## 3. Results

### 3.1. Floristic Analysis of C. ambigua Populations

Pop 1: Sultan Epe (Tyubkaragan Peninsula). The Sultan Epe Gorge is a coastal canyon with a plateau reaching about 200 m. Soils are loamy, sandy loam, gravel, solonetz, and solonchak, with eroded soils on slopes. The vegetation is heterogeneous. Dominant species are *C. ambigua* and *Morus alba*. The tree layer (3–5 m) includes *Morus nigra*, *Salix alba*, and *Elaeagnus angustifolia*. The shrub layer (40–50% cover) is dominated by *C. grandiflora* and *Rhamnus sintenesii*. Herbaceous cover includes *Cardaria draba*, *Chorispora tenella*, *Poa bulbosa*, and *Eremopyrum triticeum*. Leading families: Brassicaceae (seven species), Poaceae (four species), Asteraceae (three species), and Chenopodiaceae (three species). Life forms comprise trees (11.6%), shrubs (4.7%), subshrubs (14%), perennial herbs (30.2%), and annuals (39.5%).

Pop 2: Karakozaiym (Tyubkaragan Peninsula). The Karakozaiym Gorge, 25 km west of Fort Shevchenko, has eroded limestone-shell rocks and sandstone ridges. Vegetation includes a forb–hawthorn (*Crataegus ambigua*—*Herba varia*) community with 45–55% coverage. Dominant species are *C. ambigua* and *Morus alba*. The shrub layer includes *Rhamnus sintenesii* and *Caragana grandiflora*, with *Rubus caesius* and *Reaumuria fruticosa* near the springs. Herbaceous cover includes *Onopordum acanthium*, *Allium sabulosum*, and *Haplophyllum obtusifolim*. Leading families are Brassicaceae (seven species), Poaceae (four species), Boraginaceae (three species), Lamiaceae (three species), Rosaceae (three species), Asteraceae (two species), and Chenopodiaceae (two species). Life forms are trees (4.9%), shrubs (9.8%), subshrubs (9.8%), perennial herbs (36.5%), and annuals (39%).

Pop 3: Emdikorgan (Northern Aktau Ridge). The Emdikorgan Gorge is a small gorge (300–350 m long, 50 m wide) with steep rocky slopes. Soils are thin, gravelly, and light loam. Dominant species are *C. ambigua* and *Rhamnus sintenesii*. The shrub layer includes *Tamarix*. Herbaceous cover includes *Falcaria vulgaris*, *Galium aparine*, and *Haplophyllum* bungee. The hawthorns are in good condition, with high fruiting (over 65%). Species composition includes 58–60 species from 41 genera and 23 families. Leading families are Asteraceae, Poaceae, Chenopodiaceae, and Lamiaceae (65%). Life forms are herbaceous perennials (53.3%), annuals (29.3%), and woody plants (18.4%).

Pop 4: Samal (Western Karatau Ridge). The Samal Gorge is a deep, winding gorge (~2 km long and 30–100 m wide) with steep, rocky slopes. Vegetation includes an abundant stream with dense thickets of *Mentha longifolia*, *Teucrium polium*, *Marrubium vulgare*, and *Nepeta cataria* and tree/shrub vegetation of *C. ambigua* and *Rhamnus sintenesii*. Mesophytic and mesoxerophytic communities occupy the lower slopes. The upper and lower gentle slopes feature sparse wormwood communities with *Artemisia terrae-albae*, *Artemisia gurganica*, and *Tanacetum santolina*. The hawthorns are healthy. Dominant species are *C. ambigua* and *Mentha longifolia*. Species composition includes 55 species from 49 genera and 25 families. Leading families are Asteraceae, Brassicaceae, Lamiaceae, Poaceae, Rosaceae, and Fabaceae (58.2%). Life forms are herbaceous perennials (47.3%), annuals (23.6%), trees (9.1%), shrubs (9.1%), subshrubs (5.5%), and a single shrub species (1.8%). The results of the floristic analysis are presented in Table 3.

The floristic analysis of *C. ambigua* populations across different gorges in the Tyubkaragan Peninsula and the Western Karatau Ridge reveals diverse and heterogeneous plant species compositions. *C. ambigua* consistently appears as the dominant species across all studied populations, often accompanied by co-dominant species such as *Morus alba*, *Herba varia*, *Rhamnus sintenesii*, and *Mentha longifolia*.

The Sultan Epe Gorge’s floristic composition is rich and layered, with a variety of tree, shrub, and herbaceous species. The community includes significant representation from families such as Brassicaceae, Poaceae, Asteraceae, and Chenopodiaceae. The shrub layer is well-formed, covering around 40–50%, and includes secondary subshrubs like *Nanophyton erinaceum* and *Artemisia lercheana*. The herbaceous cover is well-developed and diverse, featuring species such as *Cardaria draba*, *Chorispora tenella*, and *Poa bulbosa*.

The Karakozaiym Gorge features a forb–hawthorn community with *C. ambigua* and Morus alba as dominant species. The shrub layer is characterised by *Rhamnus sintenesii* and *Caragana grandiflora*, with occasional presence of *Rubus caesius* and *Reaumuria fruticosa*. The herbaceous cover includes species like *Onopordum acanthium*, *Allium sabulosum*, and *Haplophyllum obtusifolim*. The leading plant families are Brassicaceae, Poaceae, Boraginaceae, Lamiaceae, and Rosaceae.

The Emdikorgan Gorge shows a rich species composition despite harsh and arid conditions. The dominant species, *C. ambigua*, is accompanied by *Rhamnus sintenesii* and *Tamarix* in the shrub layer. The herbaceous layer is diverse, with species such as *Falcaria vulgaris*, *Galium aparine*, and *Haplophyllum bungee*. The community’s species composition includes families like Asteraceae, Poaceae, Chenopodiaceae, and Lamiaceae, which together represent about 65% of the flora.

*C. ambigua* forms healthy and diverse plant communities alongside *Mentha longifolia* in the Samal Gorge. The vegetation includes mesophytic and xerophytic elements, with abundant species from families such as Asteraceae, Brassicaceae, Lamiaceae, Poaceae, Rosaceae, and Fabaceae. The herbaceous perennials dominate, and a balanced mix of trees, shrubs, and subshrubs exist.

Overall, the analysis highlights the adaptability and ecological significance of *C. ambigua* in various environmental conditions, contributing to the floristic diversity of the regions studied. The dominant and co-dominant species and the leading plant families provide insight into the complex interactions within these ecosystems.

### 3.2. Age Composition of C. ambigua in the Populations

Growing under different conditions, hawthorn plants can be distinguished by various indicators related to their growing season, productivity, and development. Identifying the characteristics of growth and development and the variability in individuals allows for targeted selection of forms for introduction into culture and subsequent selection.

When taking an inventory of the population, four groups of hawthorns of different ages were considered: juvenile (of root sucker or seed origin); virginal (large individuals that have not reached the generative period); young generative; and adult generative (Table 4).

The data show that the age composition of *C. ambigua* in the gorges is not homogenous. The Sultan Epe Gorge had the most significant number of specimens, followed by the Karakozayim Gorge. The Samal Gorge was in third place, and the smallest number of individuals was found in the Emdikorgan Gorge.

It is worth noting that the small number of hawthorn individuals in the Emdikorgan Gorge is due to this gorge’s small size. The size of the Sultan Epe Gorge, the largest one assessed in this study, explains the most significant number of specimens found therein.

A more critical parameter is the ratio of dominant age groups of the plants observed, which reflects the state of these populations. The predominance of young regenerative individuals in a given population indicates a young state and the possibility of further development; the dominance of middle-aged individuals indicates population stability; and older individuals indicate the degradation of the population and the prospect of extinction.

### 3.3. Morphological Structure of C. ambigua Populations

Populations of *C. ambigua* in Western Karatau (Mangistau region, Kazakhstan) are characterised as low trees or shrubs. Our study involved reference plants’ vegetative and generative organs from four populations of *C. ambigua* of different ages. We analysed virgin (V), young generative (YG), and adult generative (AG) plants. The following parameters of the vegetative organs of plants were studied: plant height, cm (PH); plant crown diameter, cm (CD); trunk height, cm (TH); trunk diameter, cm (TD); spike size, cm (SS); leaf length, cm (LL); leaf width, cm (SW); leaf petiole length, cm (LPL); and leaf area, cm (LA). The following parameters of generative organs were also studied: inflorescence diameter, cm (ID); number of flowers on one inflorescence, pcs (NFI); number of inflorescences on one branch, pcs (NIB); pedicel length, mm (PL); flower diameter, mm (FD); fruit weight (FM); fruit length (FL); fruit width (FW); number of seeds (NS); and seed mass (SM). The results for the main parameters are presented in Table 5 and Table 6.

A correlation matrix was also generated for the three age groups (Figure 3, Figure 4 and Figure 5) and the four-population factor was included.

A multivariate analysis of variance (MANOVA) was performed during the study, with results including Wilks’ lambda and Pillai trace indicators. The value of Wilks’ lambda was 0.3594, indicating that the differences between groups explain approximately 64% of the total variance. The degrees of freedom for this test were 9 and 82.9. The F value for Wilks’ lambda was 4.814 and the p value was 3.569 × 10^−5^. This low *p*-value indicates a high statistical significance of the test results, allowing us to reject the null hypothesis of equal mean values between the groups.

The Pillai trace test showed a value of 0.6995. Based on the sum of squares of canonical correlations, this indicator measures the overall effect of differences between groups. A high Pillai trace value indicates significant differences between groups. The degrees of freedom for this test were 9 and 108. The F value for the Pillai trace was 3.649 and the *p*-value was 0.0005185, which also indicates a high statistical significance of the results.

Thus, the results of both tests (Wilks’ lambda and Pillai trace) demonstrated the presence of significant differences between the groups with a high degree of confidence. The low p values in both tests confirm the statistical significance of the differences, allowing us to conclude that there are substantial differences between the studied groups. A significant positive correlation was recorded for the generative organs of young generative (YG) and adult generative (AG) plants. Plant height (PH) is a significant indicator of age characteristics in various populations. Therefore, we analysed variables by plant height via a box plot (Figure 6).

The morphological indicators analysis results in PCoA, considering two parameters (the indicators of vegetative and generative organs of three age groups from four populations), are presented below (Figure 7 and Figure 8).

PCoA grouping showed a noticeable difference in the fourth population according to the generative characteristics of the organs. The indicators of the vegetative organs did not reveal an equally significant difference.

Results from the ADONIS analysis indicate significant differences between groups in the data. The high coefficient of determination (R2) value (93.639%) indicates that most of the variation in the data is explained by differences between groups. The low P-value (0.001) confirms that these differences are statistically significant, meaning that the likelihood of these differences occurring by chance is extremely low.

To ensure a more accurate delineation of the four populations, we constructed dendrograms according to two parameters (vegetative and generative organs) using the UPGMA method (Figure 9).

The comparative figure shows two dendrograms of vegetative and generative traits. The differences between the fourth and the remaining populations were significant, as were the differences in the data of the vegetative and generative organs.

### 3.4. Genetic Analysis of C. ambigua Populations

Sequencing results were obtained using six genetic markers: atpF–atpH, ITS, matK, psbK–psbI, rbcL, and trnH–psbA. The raw sequence data are attached to this article in additional files in FASTA format (Appendix A). Table 7 presents the statistical parameters of the sequences (length, conserved sites, variable sites, parsimony informative sites, and singletons).

At 193 bp, the shortest sequence was recorded in psbK–psbI. The most extended sequence, at 570 bp, was identified in matK. Table 8 shows the frequency of the presence of nucleotides in the four populations for the six primary markers.

We then assessed the informativeness and usefulness of specific genetic markers for dating the genetic diversity of the studied populations. Figure 10 shows the results of constructing six UPGMA dendrograms for various genetic markers. Various figures show different phylogenetic dendrograms, giving different scenarios for the relationships of the studied populations. A unique pattern was also observed when dendrograms were constructed.

The image shows six phylogenetic trees that show genetic differences between populations of *C. ambigua*. Each tree is built on the basis of different genetic markers: Tree A is based on the matK gene, tree B is based on the trnH–psbA gene, tree C is based on the atpF–atpH gene, tree D is based on the rbcL gene, tree E is based on ITS gene, and tree F is based on the psbK–psbI gene. Each phylogenetic tree contains four populations of *C. ambigua* (Pop 1, Pop 2, Pop 3, and Pop 4). Some trees indicate the values of supporting branches (for example, 100 or 99), which means the degree of confidence in a given branch (bootstrap support).

Tree A (matK): The populations are divided into two main branches, where Pop 3 and Pop 4 have the most significant differences. Tree B (trnH–psbA): The populations are divided into two main branches, with Pop 2 and Pop 4 being the most divergent. Tree C (atpF–atpH): The populations are divided into two main branches, with Pop 1 and Pop 4 being the most divergent. Tree D (rbcL): Populations are divided similarly but with a more apparent distinction between Pop 3 and Pop 4. Tree E (ITS): This tree shows the most significant similarity between all populations. Tree F (psbK–psbI): Populations are divided into two main branches, where Pop 2 and Pop 4 have the most important differences. These phylogenetic trees help in assessing genetic differentiation and diversity between different populations of *C. ambigua* based on other genetic markers.

The results show that Robinson–Foulds distances between pairs of trees are zero in most cases, with the exception of trees constructed from psbK–psbI sequences, which show distances of 2. This means that most trees are very similar to each other, except those built from psbK–psbI, which are slightly different from the others. The results are presented in Figure 11.

Higher Shannon index values indicate greater diversity and a more even distribution of nucleotides. The Shannon index values range from 1.192 to 1.406, indicating moderately high nucleotide diversity in the sequences of each group. The Simpson index measures the probability that two randomly selected nucleotides will be the same. Higher Simpson index values indicate greater diversity, as the likelihood of selecting two identical nucleotides is lower. The Simpson index values range from 0.650 to 0.749, which also indicates moderate to high nucleotide diversity in the sequences of each group. Now, let us analyse the results for each group:matK: Shannon index: 1.334, Simpson index: 0.724. Moderately high nucleotide diversity.psbK–psbI: Shannon index: 1.303, Simpson index: 0.695. Moderate nucleotide diversity, slightly lower than matK.rbcL: Shannon index: 1.406, Simpson index: 0.749, The highest nucleotide diversity among all groups.trnH–psbA: Shannon index: 1.192, Simpson index: 0.650, The lowest nucleotide diversity among all groups.atpF–atpH: Shannon index: 1.345, Simpson index: 0.710, Moderately high nucleotide diversity.ITS: Shannon index: 1.394, Simpson index: 0.739. High nucleotide diversity, comparable to rbcL.

All sequence groups demonstrate moderate to high nucleotide diversity, indicating significant genetic diversity within the species *C. ambigua*. The rbcL and ITS groups exhibit the highest diversity, which may suggest high variability in these genes or DNA regions. The trnH–psbA group shows the lowest diversity, indicating a more conserved nature of this DNA region. These results can be helpful in understanding the genetic structure and variability of *C. ambigua* populations and for further research into this species’ evolutionary processes.

## 4. Discussion

Research on the status of *C. ambigua* populations using morphological and phylogenetic methods concerning Kazakhstan’s territory and the wider world has yet to be shared. Populations of *C. ambigua* have characteristic traits manifested in morphological and genetic states. Studies confirm the significance of differences between populations in vegetative and generative parameters of organs and the importance of genetic studies [67,68].

Despite the harsh and arid conditions and the lack of available moisture throughout the growing season, the number of species in the gorge of population 3, Emdikorgan, is higher than that in others in Western Karatau. Among the identified species, the highest proportion belongs to the families Asteraceae, Poaceae, Chenopodiaceae, and Lamiaceae, making up about 65%. The remaining families contain one to two genera and species. Population 4, Samal, includes 55 species from 49 genera and 25 families. The highest proportion of species belong to the families Asteraceae, Brassicaceae, Lamiaceae, Poaceae, Rosaceae, and Fabaceae. Species from these six leading families comprise 58.2% of the gorge’s total plant species. Analysis of the nature of life forms showed that herbaceous perennials dominate in the Samal Gorge, with 26 species (47.3% of the total number of species); in second place are herbaceous young plants, with 13 species (23.6%); and third place is occupied by trees and shrubs, with five species each (18.2%) (further encompassing three subshrubs (5.5%), two subshrubs (3.6%), one shrub species (1.8%)). Such an abundance of species diversity has also been recorded in this territory over a long period, confirming these plant communities’ stability [69].

The ages of these *C. ambigua* populations vary and are not homogeneous. In the Samal population, generative individuals of dubious hawthorn dominated. The share of generative specimens was 73.2% and of regenerative specimens was 26.8%. Thus, the population of dubious hawthorns in the Samal Gorge can be characterised as middle-aged and stable, developing with a predominance of generative individuals. In the population of Sultan Epe, there was a predominance of generative plants of dubious hawthorn. The share of young generative individuals was 42.2% of the total specimens and the proportion of adult generative individuals was 28.2%. Young and virgin individuals accounted for about 30% of the total population. Thus, the population of dubious hawthorns in the Sultan Epe Gorge can be characterised as middle-aged and developing, with a predominance of generative individuals. In the Karakozayim Gorge, two age groups dominated the dubious hawthorn population. The first age group was young generative plants (67.7%), and the second was young plants (26.9%). At the same time, the proportions of virginal plants and adult generative plants remained very low. Such a ratio is formed in populations that have experienced highly unfavourable conditions. Part of this population was destroyed, probably due to many years of drought, but it is currently recovering. Thus, we can characterise the population of dubious hawthorns in the Karakozayim Gorge as unstable and dwindling. In the Emdikorgan population, generative individuals of dubious hawthorn notably dominated. The share of generative specimens was 52.7%, and that of regenerative specimens was 47.3%; different age groups of plants were evenly represented. Thus, the hawthorn population in the Emdikorgan Gorge can be characterised as middle-aged, stable, and developing a dominant proportion of young generative plants. This structure is also consistent with that reported in a previous study [70].

The morphological parameters of vegetative and generative organs exhibited significant differences between *C. ambigua* populations. The fourth population in Samal differed from other populations in coordinates and formed an auxiliary group according to the generative trait, as presented in Figure 8. Similar studies have also reported these results, which can be used to identify critical populations of *C. ambigua* [71].

Polymerase chain reaction modes were developed for the above primers, considering their incomplete specificity for target sequences. Even though the primers used in the work are universal for most plants, they are not strictly specific for the plant species under study. The specificity of primers ranges from 90% to complete specificity of the oligonucleotide for the target. Primer annealing was optimised in this connection with a temperature gradient from 55 °C to 62 °C. To perform polymerase chain reaction (PCR) tests, the following concentrations of substances were used: template DNA—120 ng; direct primer—0.5 µM; reverse primer—0.5 µM; dNTP—0.5 mM; 1X Pfu buffer; Pfu DNA polymerase—5 activity units; and ultrapure water up to a volume of 40 µL. The amplification program for the Eppendorf Nexus Gradient thermal cycler (Hamburg, Germany) had the following temperature conditions: +98 °C (1 min)—1 cycle; +98 °C (30 s), +55–62 °C (1 min), +68 °C (2 min)—25 cycles; +68 °C (10 min)—1 cycle; and +4 °C (15 min)—1 cycle. The amplification products were separated using DNA electrophoresis in a 1.5% agarose gel in TBE buffer with ethidium bromide (10 μg/mL). The results of PCR amplification using primers for the matK, rbcL, trnH–psbA, and ITS loci were on a temperature gradient from 55 °C to 62 °C. For the psbI and atpF–atpH loci, the temperature was optimized for primer annealing.

Sequencing results using six genetic markers—atpF–atpH, ITS, matK, psbK–psbI, rbcL, and trnH–psbA—showed positive results. Genetic differences between populations were identified for four out of six markers, which indicates the robust utility of these genetic markers in populations of *C. ambigua*. Genetic markers are among the best tools for identifying key populations [46,72]. The fourth population of Samal, based on the genetic marker psbK–psbI, stood out as the root group of the others. The results of this study can be used to preserve the region’s biological diversity [73,74].

Key populations refer to specific groups within a species that are critical for the overall genetic diversity and evolutionary potential of that species. These populations often exhibit unique genetic traits or adaptations that are important for the species’ survival and resilience. In conservation biology, identifying key populations is crucial for prioritising efforts to preserve biodiversity. Genetic markers like psbK–psbI help identify these critical groups for the hawthorn populations discussed by highlighting genetic variations and relationships among the different populations.

The root group refers to the ancestral or original population from which other populations have diverged. In a phylogenetic context, the most genetically distinct group serves as the starting point for the evolutionary tree. For the Samal population of hawthorn, being identified as the root group means it is considered the ancestral source or the most genetically distinct population from which the other three populations have evolved or diversified. This identification can help understand the evolutionary history and genetic relationships among the populations, essential for effective conservation strategies.

In the study, the Samal population’s genetic marker psbK–psbI positioned it as the root group, suggesting that it has unique genetic characteristics that differentiate it from the other populations. This distinction highlights its importance in maintaining the genetic diversity and overall health of the species in the region. Consequently, the results from this study can guide conservation efforts to ensure the preservation of the biological diversity in the Mangistau region.

To compare genetic and morphological parameters, it is necessary to analyse two types of data presented in the images. Dendrograms show the genetic diversity of *C. ambigua* populations based on different genetic markers. Populations (Pop 1, Pop 2, Pop 3, and Pop 4) are compared using six different markers (matK, trnH–psbA, atpF–atpH, rbcL, ITS, and psbK–psbI).

PCA scatterplots show morphological diversity among populations based on principal components (PC1 and PC2). The first graph divides plants by growth stage (young generative plants, mature generative plants, and virgin plants). The second graph shows the overall morphological variation between populations.

Comparison of genetic and morphological diversity shows that populations of *C. ambigua* have differences in genetic composition, as evidenced by different clusters on the dendrograms. For example, based on the matK marker (plot A), populations Pop 1 and Pop 2 are more similar than populations pop 3 and pop 4. Similar results are observed for other markers.

PCA scatterplots show that morphological characteristics also vary among populations. The first graph shows young generative plants have greater morphological diversity than adult generative plants. The second graph shows that the pop 4 population has the most pronounced morphological differences compared to other populations.

The correlation between genetic and morphological data overlaps. The genetic diversity observed in the dendrograms overlaps with the morphological differences observed in the PCA plots. For example, the pop 4 population exhibits significant variation in both genetic markers and morphological characteristics.

The genetic data support the morphological observations, indicating that genetic differences may account for morphological variation among *C. ambigua* populations. Thus, the analysis shows that genetic and morphological indicators are essential for understanding the biological diversity of *C. ambigua* populations.

## 5. Conclusions

Our research on *C. ambigua* populations in the Western Karatu region of the Mangystau region of Kazakhstan has produced significant results. The positive consequences of utilising morphological analysis and genetic dating methods have been proven. Additionally, the data we obtained (via floristic analysis) on the age distribution of *C. ambigua* populations confirm the differences between populations. Regarding age structure, the fourth population of Samal has a high concentration of adult generative plants, at 30.9%. The indicator of the generative organs of young generative and adult generative plants was also proven significant. According to these indicators, this group of plants from the fourth Samal population formed a separate group. Additionally, this population features a high proportion of young generative plants, at 42.3%. According to the results of floristic analysis, the populations in the Samal Gorge are dominated by herbaceous perennial plants, which are present in the form of 26 species (47.3% of the total number of species); in second place are herbaceous young plants, with 13 species (23.6%); and third place is occupied by trees and shrubs, with five species each (18.2%) (among which there are three subshrubs (5.5%), two subshrubs (3.6%), and one shrub (1.8%)). Therefore, we can confidently state that this territory is affluent in diverse species. The comparative dendrogram created using UPGMA shows the utility of the featured genetic markers. The psbK–psbI region can be used to judge the difference between the fourth Samal population and the other three. Based on the above conclusions made after applying methods for studying material populations, we can accurately describe the populations of Western Karatau in the Mangystau region.

## Figures and Tables

**Figure 1 plants-13-01591-f001:**
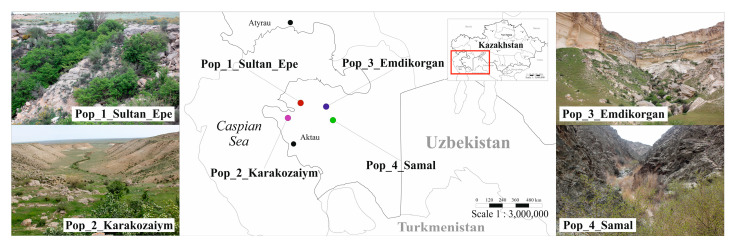
Distribution map of the four studied populations of *C. ambigua* in the Mangistau region.

**Figure 2 plants-13-01591-f002:**
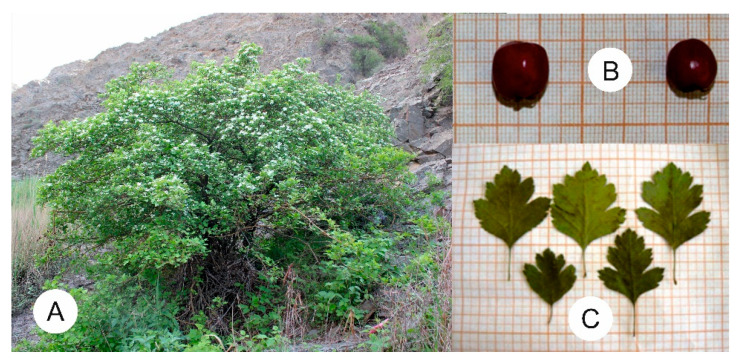
Tree *C. ambigua* adult generative five-year-old individual under natural conditions (**A**); fruits, or generative organs, and (**B**); leaves, or vegetative organs (**C**).

**Figure 3 plants-13-01591-f003:**
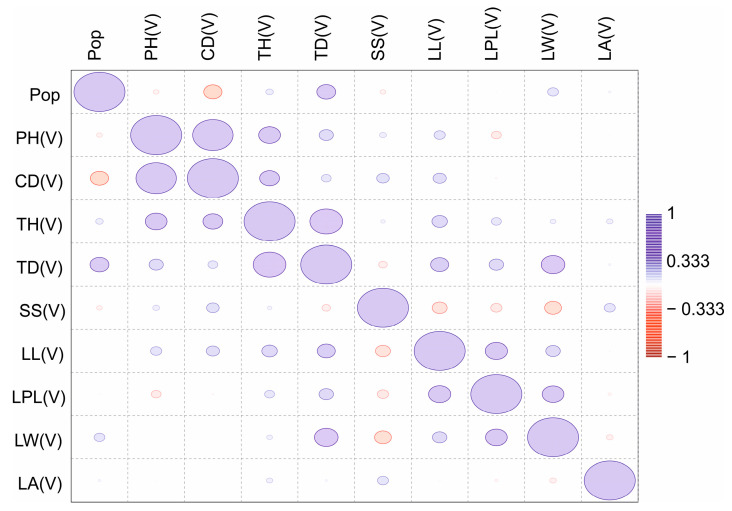
Correlation matrix of parameters of the main vegetative and generative organs of virgin (V) plants from four populations of *C. ambigua*. In the comparison matrix, from red to blue, the indicated levels of correlations are from negative to positive, and the values of the ellipses correspond to the sum of correlations.

**Figure 4 plants-13-01591-f004:**
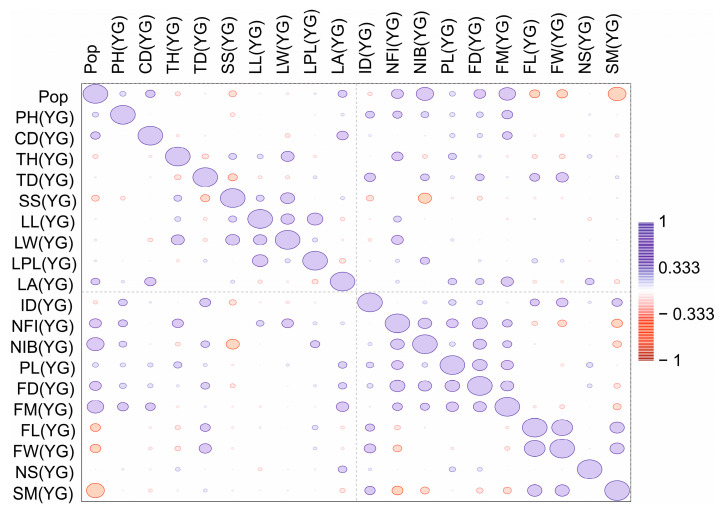
Correlation matrix of parameters of the main vegetative and generative organs of young generative (YG) plants from four populations of *C. ambigua*. In the comparison matrix, from red to blue, the indicated levels of correlations are from negative to positive, and the values of the ellipses correspond to the sum of correlations.

**Figure 5 plants-13-01591-f005:**
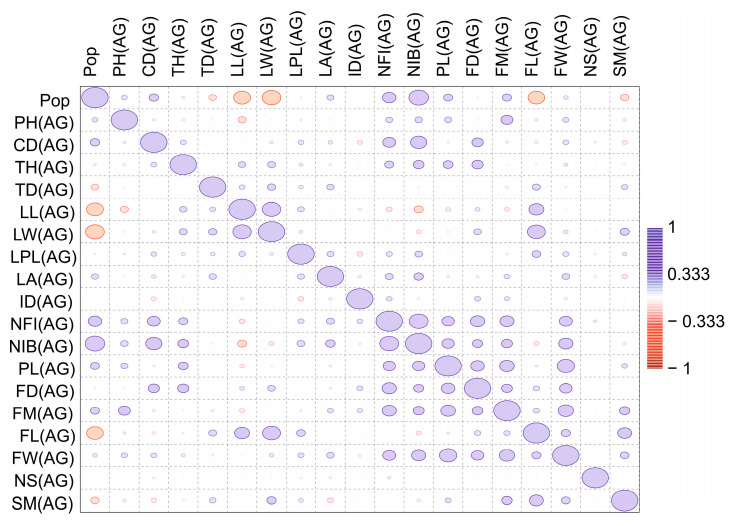
Correlation matrix of parameters of the main vegetative and generative organs of adult generative (AG) plants from four populations of *C. ambigua*. In the comparison matrix, from red to blue, the indicated levels of correlations are from negative to positive, and the values of the ellipses correspond to the sum of correlations.

**Figure 6 plants-13-01591-f006:**
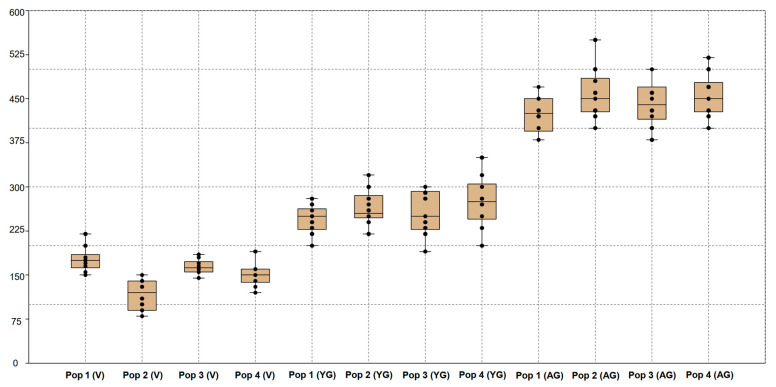
Box plot of plant height indicators from four populations of *C. ambigua* in three age groups: virgin (V), young generative (YG), and adult generative (AG). Boxes—designation of the variable frame, height indicators from the lower to the upper quartile. The lines are a designation of the maximum and minimum heights of trees. The centre line represents the middle. Black dots indicate typical and atypical parameters of tree samples.

**Figure 7 plants-13-01591-f007:**
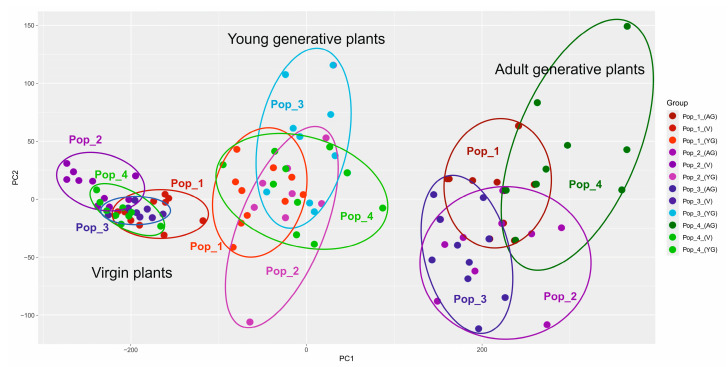
PCoA grouping of the three main age groups from four *C. ambigua* populations according to vegetative organ parameters.

**Figure 8 plants-13-01591-f008:**
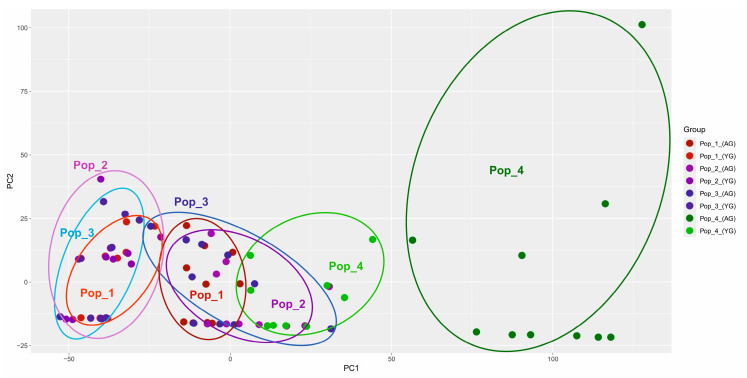
PCoA grouping of three main age groups from four populations of *C. ambigua* according to the parameters of generative organs.

**Figure 9 plants-13-01591-f009:**
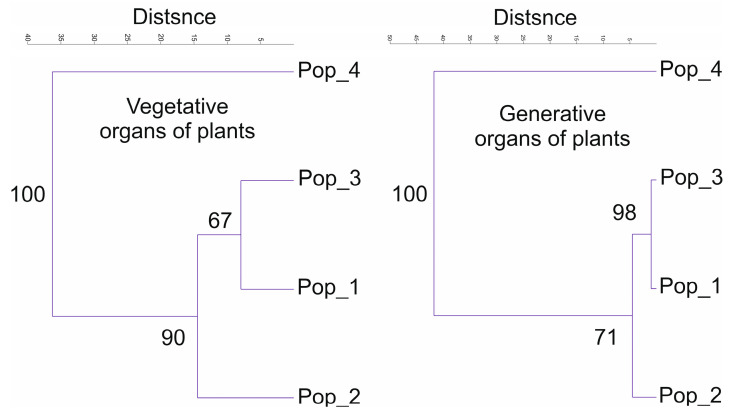
The UPGMA method was used to compare two dendrograms produced with the parameters of vegetative and generative organs of four populations of *C. ambigua*.

**Figure 10 plants-13-01591-f010:**
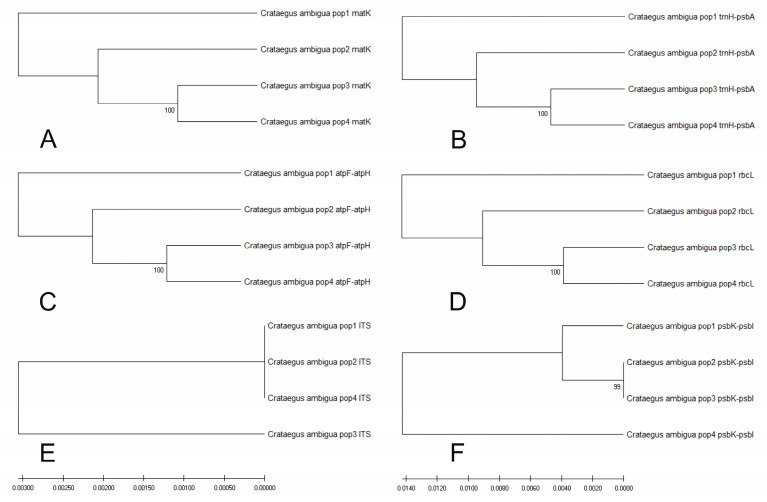
Comparative dendrograms based on UPGMA for six genetic markers: matK (**A**), trnH–psbA (**B**), atpF–atpH (**C**), rbcL (**D**), ITS (**E**), and psbK–psbI (**F**) for the studied populations of *C. ambigua*.

**Figure 11 plants-13-01591-f011:**
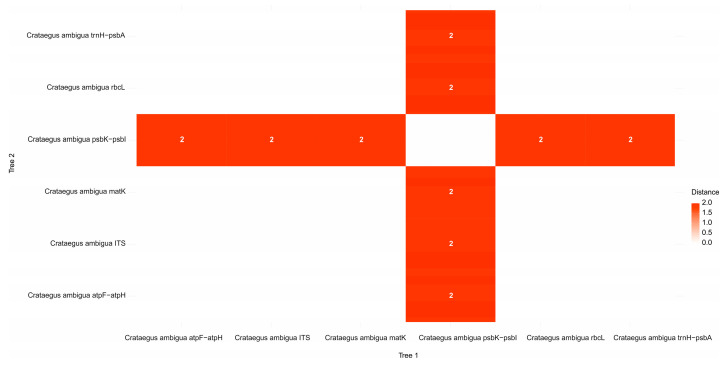
Distance matrix according to Robinson-Foulds analysis.

**Table 1 plants-13-01591-t001:** The geographical location of *C. ambigua* populations in the territory of Mangystau.

Populations	Name	Geographical Location	Coordinates	Altitude
Pop 1	Sultan Epe	Sultan Epe Gorge (Tyubkaragan Peninsula)	44°25′85.7″ N 50°58′30.7″ E	172 m
Pop 2	Karakozaiym	Karakozayim Gorge (Tyubkaragan Peninsula)	44°27′41.8″ N 50.37′71.5″ E	136 m
Pop 3	Emdikorgan	Emdikorgan Gorge (Northern Aktau Ridge)	44°28′62.8″ N 51°25′25.4″ E	35 m
Pop 4	Samal	Samal Gorge (Western Karatau Ridge)	44°07′43.6″ N 51°35′41.8″ E	247 m

**Table 2 plants-13-01591-t002:** Description of genetic primers used in the study of *C. ambigua* populations.

Name	Sequence 5′–3′	Locus for Barcode
atpF	ACTCGCACACACTCCCTTTCC	atpF–atpH
atpH	GCTTTTATGGAAGCTTTAACAAT	atpF–atpH
ITS4	TCCTCCGCTTATTGATATGC	ITS1 and ITS2
ITS5	GGAAGTAAAAGTCGTAACAAG	ITS1 and ITS2
3F_KIMf	CGTACAGTACTTTTGTGTTTACGAG	matK
1R_KIMr	ACCCCATTCATCTGGAAATCTTGGTTC	matK
psbK	TTAGCCTTTGTTTGGCAAG	psbK–psbI
psbI	AGAGTTTGAGAGTAAGCAT	psbK–psbI
rbcLa_F	ATGTCACCACAAACAGAGACTAAAGC	rbcL
rbcLa_R	GTAAAATCAAGTCCACCRCG	rbcL
psbA3f	GTTATGCATGAACGTAATGCTC	trnH–psbA
trnHf_05	CGCGCATGGTGGATTCACAATCC	trnH–psbA

**Table 3 plants-13-01591-t003:** Floristic analysis of *C. ambigua* populations.

Population	Co-Dominant Species	Tree Species	Shrub Species	Herbaceous Species
Pop 1	*Morus alba*	*Crataegus ambigua*, *Morus alba*, *Morus nigra*, *Salix alba*, *Elaeagnus angustifolia*	*Caragana grandiflora*, *Rhamnus sintenesii*, *Nanophyton erinaceum*, *Artemisia lercheana*, *Atraphaxis replicata*, *Limonium suffriticosum*, *Salsola arbuscula*	*Cardaria draba*, *Chorispora tenella*, *Psathyrostachys juncens*, *Poa bulbosa*, *Eremopyrum triticeum*, *E. orientale*, *Veronica amoema*, *Ceratocephala testiculata*, *Descurainia sophia*, *Allium sabulosum*, *Gypsophila alsinoides*, *Petrosimonia glaucescens*, *Androsace maxima*, *Scorzonera pusilla*, *Sisimbrium loeselii*, *Camelina sylvestris*, *Nonnea caspia*, *Juncus bufonius*, *Teucrium polium*, *Sinaps arvensis*, *Asparagus persicus*, *Galium aparine*, *Malva pusila*, *Marrubium vulgare*, *Equisetum ramosissima*, *Potentilla supine/*
Pop 2	*Herba varia*	*Crataegus ambigua*, *Morus alba*	*Rhamnus sintenesii*, *Caragana grandiflora*, *Rubus caesius*, *Reaumuria fruticosa*	*Onopordum acanthium*, *Allium sabulosum*, *Lagochilus acutilobus*, *Haplophyllum obtusifolim*, *Onosma staminea*, *Alopecurus arundinaceus*, *Ceratocephala testiculata*, *Crambe edentula*, *Poa bulbosa*, *Eremopyrum orientale*, *Alyssum desertorum*, *A. turkestanicum*, *Androsace maxima*, *Tragopogon ruber*, *Chorispora tenella*, *Salsola australis*, *Nonea caspica*, *Gagea peticulata*, *Lepidium ruderale*, *Bromus squarrosus*, *Minuartia regeliana*, *Fumaria parviflora*, *Veronica amoema*, *Lappula sinaica*, *Mentha longifolia*, *Potentilla transcaspia*, *Equisetum ramosissima*, *Galium humifusum*, *Cardaria draba*, *Descurainia sophia*/
Pop 3	*Rhamnus sintenesii*	*Crataegus ambigua*	*Rhamnus sintenesii*, *Tamarix*	*Falcaria vulgaris*, *Galium aparine*, *Haplophyllum bungee*, *Ixiolirion tataricum*, *Linaria leptoceras*, *Marrubium vulgare*, *Meniocus linifolius*, *Lepidium perfoliatum*, *Mentha longifolia*, *Onosma stamineum*, *Poa bulbosa*, *Prangos odontalgica*/
Pop 4	*Mentha longifolia*	*Crataegus ambigua*, *Elaeagnus angustifolia*, *Ulmus pumila*	*Crataegus ambigua*, *Rhamnus sintenesii*, *Caragana grandiflora*	*Mentha longifolia*, *Teucrium polium*, *Marrubium vulgare*, *Nepeta cataria*, *Centaurea squarossa*, *Cousinia onopordioides*, *Verbascum songaricum*, *Verbascum blattaria*, *Plantago lanceolata*, *Inula britanica*, *Medicago caerulea*, *Malva pusila*, *Equisetum ramosissimum*, *Rubus caesius*, *Stellaria media*, *Phragmites communis*, *Artemisia terrae-albae*, *Artemisia gurganica*, *Tanacetum santolina*, *Ephedra distachya*, *Echinops ritro*, *Alhagi pseudalhagi*, *Acanthophyllum pungens*, *Meristotropis triphylla*, *Stipa caspia*, *S. caucasica*, *Agropyron fragile*, *Poa bulbosa/*

**Table 4 plants-13-01591-t004:** The ratio of age groups of *C. ambigua* plants in the studied populations.

Populations	Total Copies, Pcs.	Age							
		Juveniles		Virginia		Young Generative		Adult Generative	
		Pcs.	%	Pcs.	%	Pcs.	%	Pcs.	%
Pop 1	415	89	21.4	34	8.2	175	42.2	117	28.2
Pop 2	130	35	26.9	3	2.3	88	67.7	4	3.1
Pop 3	55	15	27.27	11	20.00	22	40.00	7	12.73
Pop 4	104	16	15.3	12	11.5	44	42.3	32	30.7

**Table 5 plants-13-01591-t005:** Morphological parameters of vegetative organs of plants of four populations of *C. ambigua* (cm/cm^2^).

Morphological Parameters	Pop 1 (Mean/SD/CV)	Pop 2 (Mean/SD/CV)	Pop 3 (Mean/SD/CV)	Pop 4 (Mean/SD/CV)
Plant height (V)	185/13.2//0.07	126.7/25.2/0.20	170/10/0.06	136.7/15.3/0.1
Plant height (YG)	246.7/25.2/0.10	260/10/0.04	273.3/37.9/0.14	250/20/0.08
Plant height (AG)	400/20/0.05	426.7/5.8/0.01	450/50/0.11	456.7/11.5/0.03
Plant crown diameter (V)	126.7/14.4/0.11	80/17.3/0.22	101.7/7.6/0.08	81.7/7.6/0.09
Plant crown diameter (YG)	206.7/20.8/0.10	263.3/32.1/0.12	276.7/25.2/0.09	260/36.1/0.14
Plant crown diameter (AG)	423.3/45.1/0.11	373.3/41.6/0.11	363.3/15.3/0.04	450/50/0.11
Plant trunk height (V)	30/5/0.17	35/8.7/0.25	31/3.6/0.12	33.3/12.6/0.38
Plant trunk height (YG)	54.3/4.04/0.07	73.3/7.6/0.10	53.3/3.51/0.07	56.3/5.5/0.10
Plant trunk height (AG)	80.7/9.02/0.11	64.3/6.03/0.09	71/3.61/0.05	90/5/0.06
Plant trunk diameter (V)	10/0.5/0.05	9.3/0.58/0.06	9.5/1.32/0.14	11.7/4.16/0.36
Plant trunk diameter (YG)	15.1/0.32/0.02	12.2/1.26/0.10	13.3/1.26/0.09	15.3/1.5/0.10
Plant trunk diameter (AG)	19.3/3.51/0.18	16.3/1.53/0.09	14.2/0.76/0.05	14.3/2.08/0.15
Spike size (V)	1.07/0.21/0.20	0.87/0.15/0.18	0.9/0.1/0.11	0.9/0.1/0.11
Spike size (YG)	1.1/0.1/0.09	1.07/0.15/0.14	1.1/0.3/0.27	0.93/0.15/0.16
Leaf length (V)	3.53/0.72/0.21	3.53/0.61/0.17	3.15/0.58/0.18	3.66/0.69/0.19
Leaf length (YG)	3.46/0.68/0.20	4.15/0.76/0.18	3.07/0.48/0.16	3.67/0.60/0.16
Leaf length (AG)	5.11/0.57/0.11	4.02/0.42/0.10	3.47/0.48/0.14	3.68/0.84/0.23
Leaf width (V)	3.11/0.40/0.13	3.01/0.41/0.14	2.4/0.75/0.31	3.77/0.62/0.16
Leaf width (YG)	2.91/0.41/0.14	3.75/0.76/0.20	2.52/0.57/0.23	3.09/0.41/0.13
Leaf width (AG)	4.84/0.54/0.11	3.85/0.37/0.10	2.35/0.70/0.30	3.13/0.49/0.16
Leaf petiole length (V)	1.62/0.58/0.35	2.01/0.39/0.19	1.21/0.28/0.23	1.92/0.34/0.17
Leaf petiole length (YG)	1.7/0.45/0.27	1.6/0.39/0.25	1.38/0.23/0.17	1.91/0.54/0.28
Leaf petiole length (AG)	1.82/0.70/0.39	1.77/0.41/0.23	1.42/0.34/0.24	2.05/0.58/0.28
Leaf area (V)	9.7/1.26/0.13	10.38/1.45/0.14	10.971/1.34/0.12	9.79/1.82/0.19
Leaf area (YG)	11.1/1.53/0.14	12.81/2.37/0.19	14.95/3.15/0.21	13.21/1.93/0.15
Leaf area (AG)	13.9/1.72/0.12	14.24/3.14/0.22	13.8/2.19/0.16	15.87/1.44/0.09

**Table 6 plants-13-01591-t006:** Morphological parameters of generative organs of plants of four populations of *C. ambigua* (cm/mm/g/pcs).

Morphological Parameters	Pop 1 (Mean/SD/CV)	Pop 2 (Mean/SD/CV)	Pop 3 (Mean/SD/CV)	Pop 4 (Mean/SD/CV)
Inflorescence diameter (YG)	4.97/0.52/0.10	4.39/0.80/0.18	4.39/0.73/0.17	4.52/1.13/0.25
Inflorescence diameter (AG)	5.83/0.48/0.08	5.3/0.78/0.15	6.21/0.75/0.12	5.5/0.63/0.11
Number of flowers on one inflorescence (YG)	7.6/1.26/0.17	14.8/0.79/0.05	8.2/1.81/0.22	15.6/2.37/0.15
Number of flowers on one inflorescence (AG)	14.4/2.22/0.15	14.4/1.78/0.12	14.3/3.20/0.22	19.1/2.13/0.11
Number of inflorescences on one branch (YG)	14.4/1.71/0.12	11.3/3.71/0.33	12.2/2.86/0.23	42.8/6.94/0.16
Number of inflorescences on one branch (AG)	27.7/3.59/0.13	35.6/7.41/0.21	29.6/8.77/0.30	83.5/13.24/0.16
Pedicel length (YG)	4.46/1.00/0.22	6.35/1.97/0.31	5.49/1.12/0.20	5.92/1.76/0.30
Pedicel length (AG)	5.12/0.60/0.12	6.52/2.20/0.34	5.46/1.32/0.24	7.33/2.14/0.29
Flower diameter (YG)	13.45/1.45/0.11	14.58/1.47/0.10	14.29/1.79/0.13	15.93/1.23/0.08
Flower diameter (AG)	16.21/1.31/0.08	15.01/1.52/0.10	14.83/1.42/0.10	16.58/1.08/0.06
Fruit weight (YG)	0.672/0.20/0.30	0.943/0.21/0.23	1.102/0.17/0.16	1.122/0.14/0.13
Fruit weight (AG)	0.977/0.21/0.21	1.102/0.19/0.17	1.07/0.14/0.13	1.169/0.13/0.11
Fruit length (YG)	1.408/0.27/0.19	1.071/0.10/0.10	1.06/0.08/0.08	1.163/0.06/0.05
Fruit length (AG)	1.503/0.28/0.18	1.435/0.21/0.14	1.068/0.13/0.12	1.14/0.11/0.09
Fruit width (YG)	1.532/0.36/0.24	0.958/0.14/0.14	1.082/0.12/0.12	1.11/0.08/0.07
Fruit width (AG)	1.07/0.05/0.04	1.202/0.08/0.07	1.002/0.12/0.12	1.20/0.08/0.07
Number of seeds (YG)	1.6/0.52/0.32	1.6/0.52/0.32	1.8/0.63/0.35	1.6/0.52/0.32
Number of seeds (AG)	1.6/0.52/0.32	1.6/0.52/0.32	1.6/0.52/0.32	1.6/0.52/0.32
Seed mass (YG)	0.255/0.06/0.22	0.173/0.05/0.29	0.152/0.05/0.31	0.116/0.03/0.25
Seed mass (AG)	0.149/0.06/0.39	0.183/0.05/0.28	0.132/0.03/0.25	0.121/0.03/0.23

**Table 7 plants-13-01591-t007:** Statistical characterization of six sequences of genetic markers from *C. ambigua* populations.

Genetic Markers	Length	Conserved Sites	Variable Sites	Singletons
atpF–atpH	414	408	0	0
ITS	332	328	2	2
matK	570	570	0	0
psbK–psbI	193	185	7	6
rbcL	497	493	0	0
trnH–psbA	270	270	0	0

**Table 8 plants-13-01591-t008:** Frequency of presence of nucleotides in four populations of *C. ambigua* for six genetic markers.

Nucleotide	Pop 1	Pop 2	Pop 3	Pop 4	Avg.
atpF–atpH					
T(U)	37.7	37.3	37.5	37.5	37.5
C	12.8	12.3	12.7	12.7	12.7
A	34.0	33.9	33.8	33.8	33.9
G	15.5	16.5	15.9	15.9	16.0
Total	406	413	408	408	408.8
ITS					
T(U)	19.4	19.4	19.8	19.7	19.6
C	32.4	32.4	32.8	32.4	32.5
A	18.5	18.5	18.2	18.5	18.4
G	29.7	29.7	29.2	29.4	29.5
Total	330	330	329	330	329.8
matK					
T(U)	30.4	30.4	30.4	30.4	30.4
C	18.1	18.1	18.1	18.1	18.1
A	35.3	35.3	35.3	35.3	35.3
G	16.3	16.3	16.3	16.3	16.3
Total	570	570	570	570	570
psbK–psbI					
T(U)	43.5	43.8	44.0	44.0	43.8
C	13.5	14.1	13.6	13.6	13.7
A	27.5	27.6	27.7	27.7	27.6
G	15.5	14.6	14.7	14.7	14.9
Total	193	192	191	191	191.8
rbcL					
T(U)	29.6	29.4	29.6	29.6	29.6
C	21.3	21.1	21.3	21.3	21.3
A	26.6	27.0	26.6	26.6	26.7
G	22.5	22.5	22.5	22.5	22.5
Total	493	497	493	493	494
trnH–psbA					
T(U)	33.3	33.2	33.2	33.3	33.3
C	13.3	13.4	13.4	13.3	13.4
A	46.7	47.0	47.0	46.7	46.8
G	6.7	6.3	6.3	6.7	6.5
Total	270	268	268	270	269

Note: All frequencies are given in cents.

## Data Availability

Data are contained within the article.

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
