# Peer review of "Study of the Floristic, Morphological, and Genetic (atpF–atpH, Internal Transcribed Spacer (ITS), matK, psbK–psbI, rbcL, and trnH–psbA) Differences in *Crataegus ambigua* Populations in Mangistau (Kazakhstan)"

_plants, 2024, doi:10.3390/plants13121591_

Round 1
Reviewer 1 Report
Comments and Suggestions for Authors
Line 273:
a one-way analysis of variance (ANOVA) was used, which found a significant correlation
between soil type and the main parameters of plants in the population:
A CCA (canonical correspondence analysis) should be used for this task.
2- Results 281
Floristic analysis of C. ambigua populations
It may be presented by suitable tables indicating the species growing in that region. The text becomes much shorter and the areas studied (populations), may be compared more clearly.
In line 484:
A box plot analysis of variables by plant height was generated (Figure 6).
This should be accompanied by a proper statistical analysis.
In general for significant test of quantitative variables among the studied populations
a MANOVA (Multivariate statistical analysis of variance) is suggested.
Line 218
Genetically analysis : Should be corrected to Genetic analyses
The number of plants studied for each genetic marker should be stated and presented in clustering or dendrograms.
Data presented is based on the sequence differences of the samples and no genetic diversity indices or parameters are provided.
Line 498 (Figures 7 & 8. PCoA)
I can suggest performing the Adonis significant test for the geoups in PCoA as performed in ggpcoa package R.
Line 544 - 547
Various figures show different phylogenetic dendrograms giving different scenarios
for the relationship of the studied populations. A special pattern is also observed when
constructing dendrograms.
This is not a conclusive decision of the results. Explain in more detail.
Moreover, you can present the tree’s difference by using a proper distance value, for example, “Robinson–Foulds”.
The phylogenetic trees based on morphological and genetic data should also be compared to draw a precise conclusion.
In Discussion part:
Line 613-615:
Genetic markers are one of the main tools for identifying key populations [71-75].
The fourth population of Samal, based on the genetic marker psbK-psbI, stood out as the root group of the others. The results of this study can be used to preserve the biological diversity of the region [76-77].
Please explain and detail the meaning of key populations and the root group.
Author Response
For research article
|
Response to Reviewer 1 Comments
|
||
|
1. Summary |
|
|
|
Thank you very much for taking the time to review this manuscript. Please find the detailed responses below and the corresponding revisions/corrections highlighted/in track changes in the re-submitted files. The team of authors expresses gratitude!
|
||
|
2. Questions for General Evaluation |
Reviewer’s Evaluation |
Response and Revisions |
|
Does the introduction provide sufficient background and include all relevant references? |
Yes |
Thank you! |
|
Are all the cited references relevant to the research? |
Must be improved |
We tried to improve this! |
|
Is the research design appropriate? |
Must be improved |
We tried to improve this! |
|
Are the methods adequately described? |
Must be improved |
We tried to improve this! |
|
Are the results clearly presented? |
Must be improved |
We tried to improve this! |
|
Are the conclusions supported by the results? |
Must be improved |
We tried to improve this! |
|
3. Point-by-point response to Comments and Suggestions for Authors
|
||
|
Comments 1: Line 273: a one-way analysis of variance (ANOVA) was used, which found a significant correlation between soil type and the main parameters of plants in the population: A CCA (canonical correspondence analysis) should be used for this task.
|
||
|
Response 1: This proposal has been removed. It ended up in the manuscript by mistake. Soil analysis was not carried out in these studies.
|
||
|
Comments 2: 2- Results 281 Floristic analysis of C. ambigua populations It may be presented by suitable tables indicating the species growing in that region. The text becomes much shorter and the areas studied (populations), may be compared more clearly.
|
||
|
Response 2: This section has been revised in tabular form.
Comments 3: In line 484: A box plot analysis of variables by plant height was generated (Figure 6). This should be accompanied by a proper statistical analysis. In general for significant test of quantitative variables among the studied populations a MANOVA (Multivariate statistical analysis of variance) is suggested.
Response 3: This analysis has been added
Comments 4: Line 218 Genetically analysis : Should be corrected to Genetic analyses The number of plants studied for each genetic marker should be stated and presented in clustering or dendrograms. Data presented is based on the sequence differences of the samples and no genetic diversity indices or parameters are provided.
Response 4: This section has been added to the methodology and results.
Comments 5: Line 498 (Figures 7 & 8. PCoA) I can suggest performing the Adonis significant test for the geoups in PCoA as performed in ggpcoa package R.
Response 5:
Calculation has been added
Comments 6: Line 544 - 547 Various figures show different phylogenetic dendrograms giving different scenarios for the relationship of the studied populations. A special pattern is also observed when constructing dendrograms. This is not a conclusive decision of the results. Explain in more detail. Moreover, you can present the tree’s difference by using a proper distance value, for example, “Robinson–Foulds”. The phylogenetic trees based on morphological and genetic data should also be compared to draw a precise conclusion.
Response 6:
Calculation has been added
Comments 7: In Discussion part: Line 613-615: Genetic markers are one of the main tools for identifying key populations [71-75]. The fourth population of Samal, based on the genetic marker psbK-psbI, stood out as the root group of the others. The results of this study can be used to preserve the biological diversity of the region [76-77]. Please explain and detail the meaning of key populations and the root group.
Response 7:
Explanations have been added |
||

Reviewer 2 Report
Comments and Suggestions for Authors
The manuscript "Study of floristic, morphological, and genetic (atpF-atpH, ITS, matK, psbK-psbI, rbcL, and trnH-psbA) differences in Crataegus ambigua populations in Mangistau (Kazakhstan)" is an interesting study of a native plant species. The manuscript is potentially suitable for publication in Plants, but I suggest the authors to address major changes before I recommend the manuscript for publication.
Abstract
I consider the Abstract satisfactory, with the exception of the keywords. Keywords are used for indexing, and we usually avoid repeating the words used in the title.
Introduction
I think the Introduction needs a substantial rework. The idea of the introduction is to "introduce" the scientific question, and "introduce", based on the literature, the approach which the authors used to solve the scientific question. The authors detailed with prolixity the morphological aspects, distribution, and origin of the species, and although this information is important in a study of morphological divergence, I think there is too much details in this section (L34 - L94). Repeating the published morphological characterization of the species would be more suitable in a review article or (just lightly) in the discussion of this manuscript.
When describing the approach used to solve the scientific question (L101 - L119), the authors again made use of excessive details to present information that could be summarized in two sentences.
Methods
The item 3.1 and 3.2 were good additions to the manuscript, and they have sufficient information.
The item 3.3 has a few issues.
L218: "Genetically analysis" -> Genetic analysis
L218: How many samples per population were collected for the genetic studies?
L238-240: This seems to be more suitable in Discussion or as a separate Suplementar Material about "Primer Optimization"
The item 3.4 needs extensive review. The paragraph is vague, and the authors used jargons uncommonly applied to these statistical analysis. It will not be possible to list every case, but as one example, in the first sentence:
"Statistical processing of the results was carried out using Statistics 10 program and the capabilities of the Microsoft Excel 10.1 program". The statistical analysis are performed over the Data, the Results are the "results" of the Statistical Analysis. The two softwares presented in this sentence could be listed in a more specific way. Which methods were performed in Statistica and Excel? The authors emphasized the software, but I suggest they should emphasize the analytical methods. The softwares are just tools to perform the analytical methods.
L275: R Studio is an IDE, the actual software is R.
Results
Here the authors should present their results in a concise way. Prolixity could be observed in this section, too. Species names should be italicized (L302-L304). When presenting the results per population (L283, L316, L350, and L382), the authors repeat the bio-geographic characteristics of the ecosystem, aspects that were already addressed in the Introduction and Methods.
The results should be related to the study the authors performed. A detailed discussion of the results should be limited to the Discussion section. Font sizes are smalls in the Figures and Tables related to the correlograms (L467 - 473).
PCoA usually refer to Principal Coordinate Analysis. Also, the line 492 should not be included as a result, since it refers to a method.
L517: Are not those numbers bootstrap values? They represent the consistency of a node in the dendrogram, they are not 100 values showing differences between these populations.
The item 3.4, despite minor flaws, is the closest to what you can expect of Results section. The text is more concise, and the results were obtained from the authors experiments. Tables were good additions, but Figures are in low resolution. The sentence between the lines 544-546 is vague, and does not contribute to the results.
Discussion
The Discussion needs extensive rework. The authors repeated extensively the Results, and they limited the comparison of their results with similar studies. It will not be possible to list every case in this review, but as an example, L555-L579 is a repetition of the results presented in the previous section (item 3.1).
Comments on the Quality of English LanguageThe English language is okay, but the manuscript lacks conciseness in some sections.
Author Response
For research article
|
Response to Reviewer 2 Comments
|
|||||||||
|
1. Summary |
|
|
|||||||
|
Thank you very much for taking the time to review this manuscript. Please find the detailed responses below and the corresponding revisions/corrections highlighted/in track changes in the re-submitted files. The team of authors expresses gratitude!
|
|||||||||
|
2. Questions for General Evaluation |
Reviewer’s Evaluation |
Response and Revisions |
|||||||
|
Does the introduction provide sufficient background and include all relevant references? |
Must be improved |
We tried to improve this! |
|||||||
|
Are all the cited references relevant to the research? |
Can be improved |
We tried to improve this! |
|||||||
|
Is the research design appropriate? |
Must be improved |
We tried to improve this! |
|||||||
|
Are the methods adequately described? |
Must be improved |
We tried to improve this! |
|||||||
|
Are the results clearly presented? |
Must be improved |
We tried to improve this! |
|||||||
|
Are the conclusions supported by the results? |
Can be improved |
We tried to improve this! |
|||||||
|
3. Point-by-point response to Comments and Suggestions for Authors
|
|||||||||
|
Comments 1: Line 273: a one-way analysis of variance (ANOVA) was used, which found a significant correlation between soil type and the main parameters of plants in the population: A CCA (canonical correspondence analysis) should be used for this task.
|
|||||||||
|
|||||||||
|
|
|||||||||
|
|
|||||||||

Round 2
Reviewer 1 Report
Comments and Suggestions for Authors
The revision made is good.
Author Response
Thank you very much! We tried to improve the content and correspond to the level of a scientific journal! Your comments have helped us improve our work!
Reviewer 2 Report
Comments and Suggestions for Authors
The manuscript improved compared to the last version. However, I would recommend the authors to perform another round of reviews to their manuscript. The authors did not address the conciseness of the manuscript, and there is still unnecessary information to the text. The keywords are still repeating the words present in the title, with the exception of a single good addition to the set. Keywords present in the title add no value to the indexation of the manuscript. Figures and some tables are in very low quality, and I believe the authors can provide an improved version of these elements (e.g., Figures 3, 4, 5, and 6). Several sections in the results are more suitable to methods, so I recommend the authors to review their results to move text sections similar to the following example "The calculated Shannon and Simpson indices for each group of C. ambigua sequences allow us to assess the nucleotide diversity in each population. The Shannon index considers both the richness (number of unique nucleotides) and the evenness of their distribution. Higher Shannon index values indicate greater diversity and a more even distribution of nucleotides." to the proper section (e.g., methods/discussion). The results section should be limited to the results achieved during the author's research. The discussion improved substantially compared to the previous version, but still lacks grammar and spelling reviews.
Comments on the Quality of English Language
The English is understandable, but still needs reviews to meet the requirements of a sound scientific report in reputable journals. I recommend the authors to perform another round of grammar and spelling reviews.
Author Response
The author team expresses gratitude. We are very glad that you found flaws in our article. We tried to fix everything, and you helped us improve our work. Thank you!
